# Cluster Aware Graph Anomaly Detection

## Abstract

Graph anomaly detection has gained significant attention across various domains, particularly in critical applications like fraud detection in e-commerce platforms and insider threat detection in cybersecurity. Usually, these data are composed of multiple types (e.g., user information and transaction records for financial data), thus exhibiting view heterogeneity. However, in the era of big data, the heterogeneity of views and the lack of label information pose substantial challenges to traditional approaches. Existing unsupervised graph anomaly detection methods often struggle with high-dimensionality issues, rely on strong assumptions about graph structures or fail to handle complex multi-view graphs. To address these challenges, we propose a cluster aware multi-view graph anomaly detection method, called CARE. Our approach captures both local and global node affinities by augmenting the graph's adjacency matrix with the pseudo-label (i.e., soft membership assignments) without any strong assumption about the graph. To mitigate potential biases from the pseudo-label, we introduce a similarity-guided loss. Theoretically, we show that the proposed similarity-guided loss is a variant of contrastive learning loss, and we present how this loss alleviates the bias introduced by pseudo-label with the connection to graph spectral clustering. Experimental results on several datasets demonstrate the effectiveness and efficiency of our proposed framework. Specifically, CARE outperforms the second-best competitors by more than 39% on the Amazon dataset with respect to AUPRC and 18.7% on the YelpChi dataset with respect to AUROC. The code of our method is available at the anonymous GitHub link: https://anonymous.4open.science/r/CARE-demo-1C7F.

## 1 Introduction

Graph-based anomaly detection has been an important research area across diverse domains for decades, particularly within high-impact applications, such as fraud detection within e-commerce platforms [50, 53, 61] and insider threat detection in the cybersecurity domain [9, 19, 28]. For instance, in the realm of e-commerce, leveraging a graph-based anomaly detection algorithm proves invaluable for identifying fraudulent sellers by analyzing the properties (i.e., attributes) and connections (i.e., structure) among users [30]. Similarly, in the context of insider threat detection, constructing a graph based on users' activities allows investigators to discern anomalous users in the organization by exploring the substructure of the graph [20].

*Conference acronym 'XX, June 03–05, 2018, Woodstock, NY*

© 2018 Copyright held by the owner/author(s). Publication rights licensed to ACM.
ACM ISBN 978-1-4503-XXXX-X/18/06
https://doi.org/XXXXXXX.XXXXXXX

In the era of big data, the collected data often exhibit heterogeneous views (e. g., various data) and lack labeled data. For example, in the e-commerce platform, multiple types of data can be collected for heterogeneous graph construction, including the user's shopping history, search trends, and product ratings [29]; in credit card fraud detection, the data are composed of both cardholder information and transaction records (e. g., online purchase records) [73]. However, obtaining labels is often impractical due to the expense associated with labeling services and the demand for domain-specific expertise in discerning malicious patterns [70]. This highlights the need for innovative approaches that can deal with the intricacies of heterogeneous and unlabeled datasets.

Until then, many unsupervised anomaly detection methods [11, 23, 55] have been proposed and they can be categorized into two branches, including feature reconstruction methods and self-supervised learning methods. The feature reconstruction approaches focus on minimizing the reconstruction error of node attributes or structures [2, 13, 14]. However, these feature reconstruction based methods tend to suffer from the curse of high dimensionality [17], especially for citation network where the word occurrence is extracted as the node attributes. Another direction is the self-supervised learning methods [2, 27, 38, 49, 66, 72], which aim to design a proxy task related to anomaly detection, whereas these methods tend to have strong assumptions regarding the graph structure, thus only performing well on some graphs with certain structures. For instance, TAM [49] holds the *one-class homophily assumption* that normal nodes tend to have much stronger affinity with each other than with the abnormal nodes and the authors propose to maximize the local node similarity. However, TAM fails to consider a situation where the normal nodes and their normal neighbors might come from different classes, indicating the distinct features for these connected normal nodes. A general limitation for both branches is that many of these methods [14, 26, 72] are designed for single-view graphs, suffering from the presence of multiple views.

To address these limitations, we propose a unified Cluster AwaRE graph anomaly detection method to identify anomalous nodes in multi-view graphs, named CARE. In this work, we propose to capture both local and global node affinity and design an anomaly score function to assigning higher anomaly scores to nodes that are less similar to their neighbors based on both node attributes and structural similarity. Since the raw adjacency matrix in the graph only contains the local node affinity information, we measure the global node affinity by leveraging the pseudo-label (i.e., soft membership assignments) to augment the original adjacency matrix without any strong assumption about the graph. To reduce the potential bias introduced by the pseudo-label during the optimization, we propose a similarity-guided loss that utilizes the soft assignment to build a similarity map to help the model learn robust representations. Theoretically, we analyze that the proposed similarity-guided loss is a variant of the contrastive loss, and we present how the proposed regularization mitigates the potential bias by connecting it with

graph spectral clustering. Our main contributions are summarized below:

- A novel self-supervised framework for detecting anomalies in multi-view graphs.
- A novel similarity-guided contrastive loss for learning graph contextual information and its theoretical analysis showing the connection to graph spectral clustering.
- Theoretical analysis showing the negative impact of other types of contrastive loss.
- Experimental results on six datasets demonstrating the effectiveness and efficiency of the proposed framework.

The rest of this paper is organized as follows. After briefly reviewing the related work in Section 2, we then introduce a multi-view graph anomaly detection framework in Section 3. Next, we conduct the systematic evaluation of the proposed framework on several datasets in Section 4 before we conclude the paper in Section 5.

## 2 Related Work

In this section, we briefly review the related work on clustering and anomaly detection.

### 2.1 Clustering

In the past decades, clustering methods gradually evolved from traditional shallow methods, (e. g., Non-Negative Matrix Factorization (NMF) methods [4, 52, 68] and spectral clustering methods [5, 37, 43]), to deep learning-based methods (e. g., deep NMF [56, 69], graph neural network clustering methods [8, 39, 60], autoencoder-based clustering methods [15, 24]). For example, the authors of [52] propose an NMF-based clustering method that models the intrinsic geometrical structure of the data by assuming that several neighboring points should be close in the low dimensional subspace. The work in [4] presents a semi-NMF clustering method by taking advantage of the mutual reinforcement between data reduction and clustering tasks. Additionally, [69] extends shallow NMF to a deep Semi-NMF for multi-view clustering by learning the hierarchical structure of multi-view data and maximizing the mutual information of each pair of views. Different from these methods, this paper first follows the idea of the graph pooling method [67] to get the soft assignment, aiming to capture the global node affinity information and regularize the learned representations by the similarity-guided contrastive loss.

### 2.2 Anomaly Detection

Anomaly detection has been studied for decades [22, 35, 42, 48]. The increasing demand in many domains, such as financial fraud detection, anomaly detection in cybersecurity, etc., has attracted many researchers' attention, and a variety of outstanding algorithms have been proposed, ranging from shallow algorithms [1, 10, 40, 65] to deep models [3, 12, 34, 45, 46, 74]. To name a few, [65] presents an algorithm named Outlier Pursuit, which projects the raw data to the low-dimensional subspace and identifies the corrupted points with PCA; [12] encodes observed co-occurrence in different events into a hidden space and utilizes the weighted pairwise interactions of different entity types to define the event probability. AnoGAN [3] is a generative adversarial network (GAN) based anomaly detection method, which utilizes the generator and the discriminator to

capture the normal patterns while the anomalies are detected based on the residual score and the discrimination score. HCM-A [27] designs a self-supervised learning loss by forcing the prediction of the shortest path length between pairs of nodes. ComGA [41] proposes a community-aware attributed graph anomaly detection method to detect community structure of the graph. CONAD [66] integrates human knowledge of different anomaly types via data augmentation and introduces a contrastive learning-based method for graph anomaly detection. TAM [49] proposes a scoring measure by assigning large score to the nodes that are less affiliated with their neighbors and introduce truncated affinity maximization to reduce the bias during the optimization. In contrast, this paper proposes to capture both local and global node affinity information, and we propose similarity-guided contrastive learning loss to learn robust representations and to mitigate potential bias.

## 3 Proposed CARE Framework

In this section, we present our proposed framework, CARE, for multi-view graph anomaly detection. We begin by defining the notation and then introduce cluster-aware node affinity learning alongside the similarity-guided contrastive learning loss. Next, we outline the overall objective function and the inference process for detecting anomalies. Finally, we analyze the limitations of using weakly supervised contrastive loss in our method.

### 3.1 Notation

Throughout this paper, we use regular letters to denote scalars (e. g., $\alpha$), boldface lowercase letters to denote vectors (e. g., $\boldsymbol{x}$), and boldface uppercase letters for matrices (e. g., $\boldsymbol{A}$). Given an undirected graph $\mathcal{G} = (V, E^1, ..., E^v, X^1, ..., X^v)$, our objective is to identify anomalous nodes in the graph, where $v$ represents the number of views, $\boldsymbol{V}$ consists of $n$ vertices, $\boldsymbol{E}^v$ consists of $m^v$ edges, $\boldsymbol{X}^v \in \mathbb{R}^{n \times d_v}$ denotes the feature matrix of the $v$-th view and $d_v$ is the feature dimension. For clarity, we denote $u_i$ as node $i$, $\boldsymbol{x}_i^v \in \mathbb{R}^{d_v}$ as the node attributes of $u_i$ for the $v$-th view, $\boldsymbol{h}_i \in \mathbb{R}^d$ as the embedding of node $u_i$ by any type of GNNs and $d$ is the feature dimensionality of the hidden representation. $\boldsymbol{H}^v \in \mathbb{R}^{n \times d^v}$ is the node embedding matrix. We let $\boldsymbol{A}^v \in \mathbb{R}^{n \times n}$ denote the adjacency matrix of the $v$-th view where $A_{ij}^v = 1$ iff node $u_i$ and node $u_j$ are connected, $\boldsymbol{D}^v \in \mathbb{R}^{n \times n}$ denotes the diagonal matrix of vertex degrees for the $v$-th view, and $\boldsymbol{I} \in \mathbb{R}^{n \times n}$ denotes the identity matrix. The symbols are summarized in Table 1.

### 3.2 Cluster-Aware Node Affinity Learning

Many self-supervised learning methods [2, 27, 38, 49, 66, 72] aim to design a proxy task relevant to the anomaly detection. One branch [16, 49] is to maximize the local node similarity for a multi-view graph as follows:

$$\mathcal{L}_1 = \sum_{a=1}^{v} \sum_{u_i} \frac{1}{|\mathcal{N}^a(i)|} \sum_{u_j \in \mathcal{N}^a(i)} \text{sim}(\boldsymbol{h}_i^a, \boldsymbol{h}_j^a)$$

$$= \sum_{a=1}^{v} \sum_{u_i} \sum_{u_j} \frac{1}{D_i^a} A_{ij}^a \cdot \text{sim}(\boldsymbol{h}_i^a, \boldsymbol{h}_j^a) \quad (1)$$

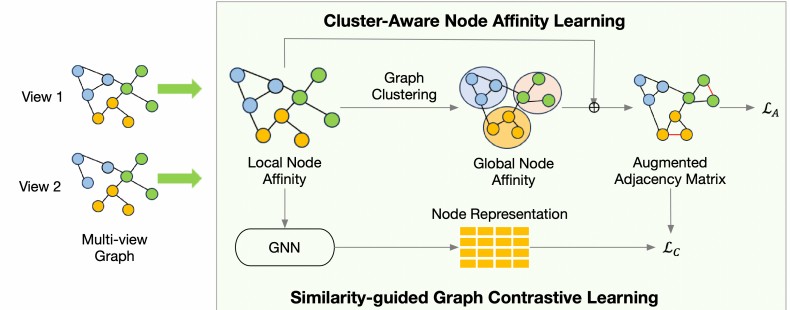

**Figure 1: The overview of CARE. It first extracts the global node affinity based on the soft assignment by graph clustering method, and then combines the global node affinity and local node affinity together. Similarity-guided graph contrastive loss is then introduced to mitigate the potential bias.**

**Table 1: Definition of Symbols**

| Symbols | Definition |
|---------|------------|
| $V$ | The set of vertices |
| $E^a$ | The set of edges for the $a$-th view |
| $A^a$ | The $a$-th view adjacency matrix |
| $X^a$ | The $a$-th view node attribute matrix |
| $v(n)$ | The number of views (nodes) |
| $D^a$ | The degree matrix for the $a$-th view |
| $H^a$ | The the $a$-th view node representations |
| $\hat{A}$ | The adjacency matrix augmented by cluster similarity |
| $M^a$ | The soft membership matrix for the $a$-th view |
| $\bar{H}$ | The average node representations |
| $\bar{M}$ | The average soft membership matrix |
| $\tilde{A}$ | The normalized augmented adjacency matrix |

where $\text{sim}(h_i^a, h_j^a) = \frac{h_i^a (h_j^a)^T}{|h_i^a||h_j^a|}$ measures the similarity of node embedding between the node $u_i$ and its neighbor $u_j$ based on the $a$-th view of the graph, $\mathcal{N}^a(i)$ denotes the neighbors of the node $u_i$ and $D_i^a$ is the degree of the node $u_i$ for the $a$-th view. However, these methods typically rely on *one-class homophily assumption*, which posits that normal nodes tend to exhibit strong affinities with each other, while the affinities among abnormal nodes are significantly weaker. This assumption is overly restrictive, as it focuses exclusively on extracting local node affinities while neglecting global node affinities.

A naive solution to relax this constraint is to incorporate high-order information by extending the local 1-hop neighbors to $k$-hop neighbors. However, this approach presents two main issues. First, including high-order neighbors inevitably introduces more abnormal nodes during the node affinity maximization process, resulting in a sub-optimal solution. Second, it overlooks a crucial scenario where normal anchor nodes and their neighbors may belong to different classes, indicating that these connected normal nodes may have distinct features. Since both normal neighbors from different classes and abnormal neighbors possess features that differ from those of normal anchor nodes, incorporating these distinct features in node affinity learning decreases the likelihood of detecting abnormal nodes. Furthermore, adding high-order neighbors in node affinity learning also increases the probability of including neighbors from different classes, exacerbating the problem. To address this issue, we propose incorporating label information into the local

node affinity maximization as follows:

$$\mathcal{L}_2 = \sum_{a=1}^{v} \sum_{u_i} \sum_{u_j} \frac{1}{\sum_{u_j}(A_{ij}^a + S_{ij}^a)} (A_{ij}^a + S_{ij}^a) \cdot \text{sim}(h_i^a, h_j^a) \quad (2)$$

where $S_{ij} = 1$ if node $u_i$ and node $u_j$ belong to the same class and $S_{ij} = 0$ otherwise. Compared to $\mathcal{L}_1$, $\mathcal{L}_2$ further encodes the label information in the node affinity learning. However, label information is often unavailable due to the high costs of labeling services and the rapid growth of new data. To address this issue, we propose replacing the unavailable label information with pseudo-labels derived from a graph clustering method. Following the idea of differential graph pooling [67], we employ a one-layer Graph Convolutional Network (GCN) [32] with a softmax activation function to model soft membership assignments as follows:

$$M^a = \text{GCN}^a(A^a, X^a, W^a)$$
$$\bar{M} = \frac{1}{v} \sum_{a=1}^{v} M^a \quad (3)$$

where $W^a \in \mathbb{R}^{d_a \times c}$ is the weight matrix of GCN for the $a$-th view and $c$ is the number of clusters and we aggregate soft membership assignments from all views to obtain $\bar{M}$. We then augment the adjacency matrix by incorporating the clustering results as follows:

$$\hat{A} = (1 - \alpha)\bar{A} + \alpha \bar{M}\bar{M}^T \quad (4)$$

where $\bar{A} = \frac{1}{v} \sum_{a=1}^{v} A^a$ and $\alpha \in [0, 1]$ is a hyper-parameter balancing the importance between raw adjacency matrix and the similarity of the soft membership assignments. Our goal is to maximize the cluster-aware node affinity as follows:

$$\mathcal{L}_A(u_i) = \sum_{u_j} \frac{1}{\hat{D}_i} \hat{A}_{ij} \cdot \text{sim}(\bar{h}_i, \bar{h}_j) + \sum_{a=1}^{v} ||h_i^a - \bar{h}_i||_2 \quad (5)$$

where $\hat{D}_i = \sum_{u_j} \hat{A}_{ij}$ measures the degree of node $u_i$ in the new adjacency matrix and $\bar{h}_i = \frac{1}{v} \sum_{a=1}^{v} h_i^a$ is the average node representation. The second term enforces consistency in node embeddings across all views. However, optimization using Eq. 5 may be significantly biased by low-quality soft membership assignments at the early stages. To address this issue, we introduce the similarity-guided graph contrastive regularization.

### 3.3 Similarity-guided Graph Contrastive Regularization

To mitigate potential bias introduced by low-quality soft membership assignments, we propose a similarity-guided graph contrastive loss that minimizes the difference between the similarity of the soft assignment and the similarity of node representations for any pair of nodes. This is formulated as follows:

$$\mathcal{L}_2 = \min_{\bar{H}} ||\bar{M}\bar{M}^T - \bar{H}\bar{H}^T||_F^2 \tag{6}$$

where $\bar{M}$ is the soft assignment matrix computed in Eq. 3. $\mathcal{L}_2$ aims to learn the hidden representations such that the representations of node $u_i$ and node $u_j$ are expected to be close in the latent space if their soft membership assignments are similar. Following the idea of many existing methods restoring graph structure along with the node attributes of the graph [2, 27, 49, 72], we propose to take the graph topological structure into consideration, reformulating the similarity-guided graph contrastive loss as:

$$\mathcal{L}_C = \min_{\bar{H}} ||\tilde{A} - \bar{H}\bar{H}^T||_F^2 \tag{7}$$

where $\tilde{A} = \tilde{D}^{-1/2}\hat{A}\tilde{D}^{-1/2}$ is the normalized augmented adjacency matrix, and $\tilde{D} \in \mathbb{R}^{n \times n}$ is the diagonal matrix with $\tilde{D}_{ii} = \sum_{u_j} \hat{A}_{ij}$.

**Theoretical Justification.** We aim to provide a theoretical analysis of how the proposed similarity-guided graph contrastive regularization mitigates potential bias by connecting it with graph spectral clustering. Before delving into the direct analysis of bias mitigation, we first demonstrate that $\mathcal{L}_C$ functions as a contrastive learning loss.

**Lemma 3.1.** *(Similarity-guided Graph Contrastive Loss) Let $\bar{M}$ be the output of a one-layer graph neural network defined in Eq. 3. Then, we have*

$$\mathcal{L}_C = \mathcal{L}_f + C \tag{8}$$

*where $\mathcal{L}_f = -\sum_{i=1}^n \sum_{j=1}^n \log \frac{\exp(2\tilde{A}_{ij}\bar{h}_i\bar{h}_j^T)}{\Pi_{k=1}^n \exp((\bar{h}_i\bar{h}_k^T)^2)^{1/n}}$ is a graph contrastive loss and $C$ is a constant.*

**See proof in Appendix A.1.**

**Remark:** Compared to traditional contrastive learning losses, the denominator of $\mathcal{L}_f$ in Lemma 3.1 is the product of the exponential similarity between two node embeddings rather than a summation. The weakly supervised contrastive methods [57, 71] impose a constraint that forces a given pair of nodes to form a positive/negative pair based on their likelihood of being assigned to the same cluster (further discussion can be found in Subsection 3.5). Unlike these "discrete" formulations regarding positive and negative pairs, our approach is in a continuous form. In $\mathcal{L}_f$, $S_{ij} = \bar{M}_i\bar{M}_j^T$ can be interpreted as the similarity measurement of a node pair $(u_i, u_j)$ in terms of soft assignment, guiding the similarity of the representations of two nodes in the latent space. If we ignore the influence of the adjacency matrix in $\tilde{A} = \tilde{D}^{-1/2}\hat{A}\tilde{D}^{-1/2} = \tilde{D}^{-1/2}(\alpha\bar{M}\bar{M}^T + (1-\alpha)\frac{1}{v}\sum_{a=1}^v A)\tilde{D}^{-1/2}$ by setting $\alpha$ to be 1 (i.e., $\tilde{A} = \tilde{D}^{-1/2}\bar{M}\bar{M}^T\tilde{D}^{-1/2}$), it would be interesting to see that when two nodes are sampled from two distant clusters, $\tilde{A}_{ij} \approx 0$ and $\log \frac{\exp(2S_{ij}\bar{h}_i\bar{h}_j^T)}{\Pi_{k=1}^n \exp((\bar{h}_i\bar{h}_k^T)^2)^{\frac{1}{n}}} = 0$ for this pair of nodes. Notice that for

the node pair $(u_i, u_j)$ with high confidence being assigned to the same cluster, the weight $S_{ij}$ is larger than the weight $S_{ik}$ for the uncertain node pair $(u_i, u_k)$, where the node $u_k$ has low confidence to be assigned to the same cluster as $u_i$. By reducing the weight for these unreliable positive pairs, the negative impact of uncertain pseudo-labels can be alleviated.

Next, we aim to clarify our proposed contrastive loss by demonstrating the connection between $\mathcal{L}_C$ and graph spectral clustering.

**Lemma 3.2.** *(Graph Contrastive Spectral Clustering) Let $\bar{M}$ be the output of a one-layer graph neural network defined in Eq. 3 and $\bar{h}_i$ and $\bar{h}_j$ be unit vectors. Then, minimizing $\mathcal{L}_C$ is equivalent to minimizing the following loss function:*

$$\min \mathcal{L}_C = \min[2Tr(\bar{H}^T L\bar{H}) + R(\bar{H})] \tag{9}$$

*where $L = I - \tilde{A}$ can be considered as the normalized graph Laplacian, $I$ is the identity matrix and $R(\bar{H}) = ||\bar{H}\bar{H}^T||_F^2$ is the regularization term.*

**See proof in Appendix A.2.**

**Remark:** Based on Lemma 3.2, $\mathcal{L}_C$ can be considered as the graph spectral clustering. Graph spectral clustering [59] aims to find clusters that minimize connections between different clusters while maximizing the connections within each cluster. Traditional graph spectral clustering [6] aims to find the embedding $\bar{H}$ such that $Tr(\bar{H}^T L'\bar{H})$ is minimized, where $L'$ is the normalized graph Laplacian. The first term of Eq. 9 is similar to the objective function in traditional graph spectral clustering, but we enhance it by incorporating the similarity measurement $S_{ij} = \bar{M}_i\bar{M}_j^T$ into the normalized graph Laplacian. By including the similarity measurement in the graph spectral clustering, we reinforce $\mathcal{L}_C$ to mitigate the bias introduced by the clustering method defined in Eq. 3. It is important to note that in Lemma 3.2, the constraint that $\bar{h}_i$ and $\bar{h}_j$ are unit vectors is a common practice in many existing works [25, 62, 75]. This constraint can be easily implemented through normalization, i.e., $\bar{h}_i = \frac{\bar{h}_i}{||\bar{h}_i||_2}$.

### 3.4 Objective Function and Inference

Now, we are ready to introduce the overall objective function:

$$\min J = -\sum_{u_i} \mathcal{L}_A(u_i) + \lambda\mathcal{L}_C \tag{10}$$

where $\mathcal{L}_A$ is the cluster-aware node affinity loss and $\mathcal{L}_C$ is the similarity-guided graph contrastive loss. $\lambda$ is a constant parameter balancing these two terms. During the inference stage, we directly use the cluster-aware node affinity loss $\mathcal{L}_A$ as the abnormal score:

$$score_i = -\mathcal{L}_A(u_i) \tag{11}$$

### 3.5 Why can't weakly supervised contrastive learning loss be used as a regularization?

One effective remedy to reduce uncertainty and learn high-quality representations in an unsupervised setting is through contrastive learning loss, which has demonstrated significant performance improvements in representation quality [25, 51, 58, 70]. However, an existing study [70] has theoretically proved that simply applying vanilla contrastive learning loss (i.e., InfoNCE [58]) can easily lead to the suboptimal solution. Similarly, according to [62], both normal

**Table 2: Statistics of the datasets, including the number of nodes, anomalies, and edges for two views.**

| Name | $|V|$ | $|E^1|$ | $|E^2|$ | # Anomalies |
|---|---|---|---|---|
| BlogCatalog | 5,196 | 171,743 | - | 298 |
| Amazon | 10,244 | 175,608 | - | 445 |
| YelpChi | 24,741 | 49,315 | - | 597 |
| CERT | 1,000 | 24,213 | 22,467 | 70 |
| IMDB | 4,780 | 1,811,262 | 419,883 | 334 |
| DBLP | 4,057 | 299,499 | 520,440 | 283 |

and abnormal nodes are uniformly distributed in the unit hypersphere in the latent space by minimizing the vanilla contrastive learning loss, which leads to worse performance for the anomaly detection task. To address this issue, many weakly supervised contrastive losses [57, 71] are proposed by incorporating the semantic information, such as the clustering results, into a contrastive regularization term as follows:

$$\mathcal{L}_3 = - \sum_{i=1}^{n} \sum_{j \in C(i), j \neq i} \log \frac{\text{sim}(\bar{h}_i, \bar{h}_j)}{\text{sim}(\bar{h}_i, \bar{h}_j) + \sum_{k \notin C(i)} \text{sim}(\bar{h}_i, \bar{h}_k)} \quad (12)$$

where $\bar{h}_i = \sum_{a=1}^{v} h_i^a$ is the representation for node $u_i$ aggregated over all views, $\text{sim}(\bar{h}_i, \bar{h}_j) = \exp(\bar{h}_i \bar{h}_j^T / \tau)$ and $\tau$ is the temperature. $j \in C(i)$ means that node $u_j$ and node $u_i$ are assigned into the same cluster or form a positive pair, while $k \notin C(i)$ means that node $u_k$ and node $u_i$ are assigned into two different clusters, resulting in a negative pair. The intuition of the above equation is that if two nodes are from the same cluster, they should be close in the latent space by maximizing their similarity. However, we find out that the construction of the positive and negative pairs in Eq. 12 heavily relies on the quality of the soft assignment, while directly converting the soft assignment to the binary membership inevitably introduces bias/noise during the training phase. This bias will be amplified further in the node affinity learning as we mentioned in Subsection 3.2. (We also validate this in the ablation study in Subsection 4.3.) Here, we theoretically analyze that including this bias in the weakly supervised contrastive loss defined in Eq. 12 leads to suboptimal solution. Formally, we first define what is a true positive pair and a false positive pair respectively, and then introduce Theorem 3.4 to show the issue in Eq. 12.

**Definition 3.3.** Given a sample $x_i$, we say $(x_i, x_j)$ is a true positive pair (or a false negative pair), if their optimal representations satisfy $\exp(\bar{h}_i \bar{h}_j^T / \tau) > 1$ for a small positive value $\tau$. Similarly, we say $(x_i, x_k)$ is a false positive pair (or a true negative pair), if their optimal representations satisfy $\exp(\bar{h}_i \bar{h}_k^T / \tau) \approx 0$ for a small positive value $\tau$.

**Theorem 3.4.** *Given the contrastive learning loss function $\mathcal{L}_3$, if there exists one false positive sample in the batch during training, the contrastive learning loss will lead to a sub-optimal solution.*

**See proof in Appendix A.3.**

## 4 Experimental Results

In this section, we demonstrate the performance of our proposed framework in terms of both effectiveness and efficiency by comparing it with state-of-the-art methods.

### 4.1 Experiment Setup

*4.1.1 Datasets:* We evaluate the performance of our proposed framework on six datasets for both single-view and multi-view graph anomaly detection scenarios, including the Insider Threat Test (CERT) [20], DBLP [63], IMDB [63], BlogCatalog [54], Amazon [18] and YelpChi [33] datasets. Among these datasets, CERT, IMDB, and DBLP are multi-view graphs, while BlogCatalog, Amazon, and YelpChi are single-view graphs. CERT, Amazon, and YelpChi are real-world datasets, whereas IMDB, DBLP, and BlogCatalog are semi-synthetic graphs. (See Appendix B.1 for the details of generating anomalous nodes.) Specifically, the CERT dataset is a collection of synthetic insider threat test datasets that provides both synthetic background data and data from synthetic malicious actors. This dataset does not include a feature matrix, so we use node2vec [21] to extract two feature matrices as two views. IMDB is a movie network, where each node corresponds to a movie, and two adjacency matrices indicate whether two movies share the same actor or director. DBLP is a citation network, where each node corresponds to an academic research paper, and two adjacency matrices indicate whether two papers share the same authors or if one paper cites another. Amazon is a review network, where each node represents a product in the musical instruments category, and its attributes are extracted from product reviews. Similarly, the Yelp dataset contains hotel and restaurant reviews, either filtered (spam) or recommended (legitimate) by Yelp [18]. The statistics of these graphs are summarized in Table 2.

*4.1.2 Experiment Setting:* The neural network structure of the proposed framework is GCN [32]. The hyper-parameters $\alpha$ and $\lambda$ for each dataset are specified in Appendix B.2. In all experiments, we set the initial learning rate to be 1e-5, the hidden feature dimension to be 128 and use Adam [31] as the optimizer. The similarity function $\text{sim}(a, b)$ is defined as $\text{sim}(a, b) = \exp(\frac{a \cdot b^T}{|a||b|})$. We use TAM as the backbone of our method to capture local node affinity. The number of GCN layers is set to 2. The experiments are performed on a Windows machine with a 24GB RTX 4090 GPU. The code of our framework can be found in the anonymous GitHub link *.

*4.1.3 Evaluation Metrics:* Following [44, 49], all methods are evaluated based on Area Under the Receiver Operating Characteristic Curve (AUROC) and Area Under the Precision-Recall Curve (AUPRC). Higher AUROC/AUPRC indicates better performance. All of the experiments are repeated five times with different random seeds and the mean and standard deviation are reported.

*4.1.4 Baseline Methods:* In our experiments, we compare our proposed framework CARE with state-of-the-art methods in the following two settings.

For *single-view graphs*, we compare CARE with the following eight baseline methods: (1). **ANOMALOUS** [47]: a shallow method, jointly conducting attribute selection and anomaly detection as a whole based on CUR decomposition and residual analysis; (2). **Dominant** [14]: a graph auto-encoder-based deep neural network model for graph anomaly detection, which encodes both the topological structure and node attributes to node embedding; (3). **CoLA** [38]: a contrastive self-supervised graph anomaly detection method by

---

*https://anonymous.4open.science/r/CARE-demo-1C7F

exploiting the local information; (4). **SLGAD** [72]: an unsupervised framework for outlier detection based on unlabeled in-distribution data, which uses contrastive learning loss as a regularization; (5). **HCM-A** [27]: a self-supervised learning by forcing the prediction of the shortest path length between pairs of nodes; (6). **ComGA** [41]: a community-aware attributed graph anomaly detection framework; (7). **CONAD** [66]: a contrastive learning-based graph anomaly detection method; (8). **TAM** [49]: an unsupervised anomaly method, proposing a scoring measure by assigning large score to the nodes that are less affiliated with their neighbors.

For *multi-view graphs*, we compare CARE with the following five baseline methods: (1). **MLRA** [36]: a multi-view non-negative matrix factorization-based method for anomaly detection, which performs cross-view low-rank analysis for revealing the intrinsic structures of data; (2). **NSNMF** [2]: an NMF based method, incorporating the neighborhood structural similarity information into the NMF framework to improve the anomaly detection performance; (3). **SRLSP** [64]: a multi-view detection method based on the local similarity relation and data reconstruction; (4). **NC-MOD** [13]: an auto-encoder-based multi-view anomaly detection method, which proposes neighborhood consensus networks to encode graph neighborhood information; (5) We also report the performance of **TAM** [49] on these multi-view graphs by averaging the adjacency matrix or concatenating two feature matrices together. We omit the comparison with other baseline methods since TAM outperforms them.

## 4.2 Experimental Analysis

*4.2.1 Multi-view Graph Scenario.* In this subsection, we evaluate the performance of CARE by comparing it with five baseline methods on three multi-view graphs, i. e., CERT, IMDB and DBLP datasets. The experimental results with respect to AUROC and AUPRC are presented in Table 3. We observe the following: (i) MLRA and NSNMF exhibit the poor performance among all baseline methods across most datasets. This can be attributed to their design, which is optimized for independent and identically distributed data (e.g., image data). Thus, these two methods struggle to capture graph topological information, resulting in lower performance in both AUPRC and AUROC. (ii) In contrast, NCMOD incorporates graph topological information into the learned representation through its proposed neighborhood consensus networks, thus achieving the second-best performance on the IMDB and DBLP datasets. (iii) TAM suffers from the inability to handle multi-view graphs, performing worse than several self-supervised learning methods designed for multi-view graphs (e.g., NCMOD). (iv) Our proposed method outperforms all baselines in terms of AUROC and AUPRC across all datasets. Notably, graph-based anomaly detection methods designed for single views (e.g., TAM) falter in the presence of view heterogeneity as simply concatenating input features from multiple views distorts the feature space, resulting in significant performance challenges. In contrast, our method excels by encoding both local and global node affinities in the learned representation and mitigating biases through the proposed similarity-guided graph contrastive loss.

*4.2.2 Single-view Graph Scenario.* Next, we evaluate the performance of CARE by comparing it to eight baseline methods on three single-view graphs, i. e., BlogCatalog, Amazon and YelpChi datasets. The experimental results with respect to AUROC and AUPRC are presented in Table 4. We have the following observations: (i) TAM outperforms most baseline methods across three single-view graphs. We attribute its great performance to the design of normal structure-preserved graph truncation to remove edges connecting normal and abnormal nodes. (ii) While TAM excels on BlogCatalog, CARE still ranks as a close second, showcasing its robustness in handling single-view graphs. (iii) CARE proves to be a highly competitive approach, demonstrating superior performance on the Amazon and YelpChi datasets. Notably, CARE surpasses TAM, the second-best method, by over 15.9% in AUROC and 39.3% in AUPRC on the Amazon dataset. Similarly, on the YelpChi dataset, CARE improves AUROC by over 18.7% compared to TAM. The ability of CARE to capture both local and global node affinity information in the learned representations enables it to perform effectively on the Amazon and YelpChi datasets. By leveraging its similarity-guided graph contrastive loss, CARE enhances representation learning, allowing it to better distinguish between normal and anomalous nodes. Overall, these results emphasize that while TAM leverages truncated affinity maximization techniques to tailor the raw adjacency matrix, CARE offers a more flexible and powerful approach by incorporating global and local affinities, making it a highly effective method in the single-view graph anomaly detection scenario. Its ability to significantly outperform other baselines on challenging datasets like Amazon and YelpChi demonstrates the effectiveness of the proposed method.

## 4.3 Ablation Study

In this subsection, we conduct an ablation study to demonstrate the necessity of each component of CARE and validate the effectiveness of similarity-guided graph contrastive loss over vanilla and weakly supervised contrastive loss. Specifically, CARE-A removes the similarity measurement of soft membership and replaces $\mathcal{L}_A$ with $\mathcal{L}_2$. CARE-G refers to a variant of our proposed method by removing similarity-guided contrastive learning loss. CARE-InfoNCE replaces the similarity-guide contrastive loss with the vanilla contrastive loss while CARE-WSC substitutes it with the weakly supervised loss as shown in Eq. 12. The experimental results with respect to AUROC on the BlogCatalog, Amazon, DBLP, and IMDB datasets are presented in Table 5. Our observations are as follows:

- **Global Node Affinity**: CARE shows slight improvements on Amazon and BlogCatalog compared to CARE-A. However, excluding the global node affinity matrix (i. e., the similarity map of the soft membership) in CARE-A results in significant performance drops of approximately 40% on DBLP and 56% on IMDB, highlighting the importance of capturing global affinities in certain datasets.
- **Similarity-Guided Loss**: Removing the similarity-guided graph contrastive loss (i.e., CARE-G) leads to an average AUROC performance drop of 15% across the four datasets, underscoring the critical role of this regularization in mitigating bias and improving performance.
- **Contrastive Loss Substitutions**: Replacing the similarity-guided loss with either the vanilla contrastive loss (i.e., CARE-InfoNCE) or the weakly supervised contrastive loss

**Table 3: Results on multi-view graphs (i. e., CERT, IMDB, DBLP) with respect to AUROC and AUPRC. We boldface the best performance and underline the second-best.**

| Method | CERT | | IMDB | | DBLP | |
|---|---|---|---|---|---|---|
| | AUPRC | AUROC | AUPRC | AUROC | AUPRC | AUROC |
| MLRA | 0.0379 ± 0.001 | 0.3829 ± 0.003 | 0.2695 ± 0.007 | 0.5926 ± 0.005 | 0.2211 ± 0.005 | 0.5568 ± 0.005 |
| NSNMF | 0.0704 ± 0.001 | 0.4578 ± 0.001 | 0.0634 ± 0.000 | 0.4969 ± 0.001 | 0.1436 ± 0.007 | 0.6418 ± 0.001 |
| NCMOD | 0.0749 ± 0.001 | 0.5133 ± 0.001 | 0.6629 ± 0.013 | 0.8030 ± 0.007 | 0.4809 ± 0.006 | 0.7271 ± 0.004 |
| SRSLP | 0.0806 ± 0.007 | 0.5405 ± 0.003 | 0.5552 ± 0.017 | 0.7343 ± 0.003 | 0.0643 ± 0.002 | 0.5228 ± 0.001 |
| TAM | 0.0771 ± 0.007 | 0.5400 ± 0.005 | 0.6521 ± 0.016 | 0.8233 ± 0.013 | 0.3466 ± 0.016 | 0.6690 ± 0.005 |
| CARE | **0.1198 ± 0.003** | **0.6056 ± 0.001** | **0.8804 ± 0.024** | **0.8792 ± 0.031** | **0.6380 ± 0.027** | **0.8868 ± 0.007** |

**Table 4: Results on single-view datasets (i. e., BlogCatalog, Amazon, YelpChi) with respect to AUROC and AUPRC. We boldface the best performance and underline the second-best.**

| Method | BlogCatalog | | Amazon | | YelpChi | |
|---|---|---|---|---|---|---|
| | AUPRC | AUROC | AUPRC | AUROC | AUPRC | AUROC |
| ANOMALOUS | 0.0652 ± 0.005 | 0.5652 ± 0.025 | 0.0558 ± 0.001 | 0.4457 ± 0.005 | 0.0519 ± 0.002 | 0.4956 ± 0.003 |
| Dominant | 0.3102 ± 0.011 | 0.7590 ± 0.010 | 0.1424 ± 0.002 | 0.5996 ± 0.002 | 0.0395 ± 0.020 | 0.4133 ± 0.100 |
| CoLA | 0.3270 ± 0.000 | 0.7746 ± 0.009 | 0.0677 ± 0.001 | 0.5898 ± 0.011 | 0.0448 ± 0.002 | 0.4636 ± 0.001 |
| SLGAD | 0.3882 ± 0.007 | 0.8123 ± 0.002 | 0.0634 ± 0.005 | 0.5937 ± 0.005 | 0.0350 ± 0.000 | 0.3312 ± 0.035 |
| HCM-A | 0.3139 ± 0.001 | 0.7980 ± 0.004 | 0.0527 ± 0.015 | 0.3956 ± 0.014 | 0.0287 ± 0.012 | 0.4593 ± 0.005 |
| ComGA | 0.3293 ± 0.028 | 0.7683 ± 0.004 | 0.1153 ± 0.005 | 0.5895 ± 0.010 | 0.0423 ± 0.000 | 0.4391 ± 0.000 |
| CONAD | 0.3284 ± 0.004 | 0.7807 ± 0.003 | 0.1372 ± 0.009 | 0.6142 ± 0.008 | 0.0405 ± 0.002 | 0.4588 ± 0.003 |
| TAM | **0.4182 ± 0.225** | **0.8248 ± 0.003** | 0.2634 ± 0.008 | 0.7064 ± 0.008 | 0.0778 ± 0.009 | 0.5643 ± 0.007 |
| CARE | 0.4043 ± 0.010 | 0.8194 ± 0.003 | **0.6563 ± 0.011** | **0.8656 ± 0.002** | **0.1218 ± 0.003** | **0.7516 ± 0.003** |

**Table 5: Ablation study on BlogCatalog, Amazon, DBLP and IMDB datasets with respect to AUROC.**

| Model | BlogCatalog | Amazon | DBLP | IMDB | AVERAGE |
|---|---|---|---|---|---|
| CARE | **0.8194 ± 0.000** | **0.8656 ± 0.002** | **0.8868 ± 0.007** | **0.8792 ± 0.031** | **0.8628** |
| CARE-A | 0.8144 ± 0.000 | 0.8514 ± 0.003 | 0.4831 ± 0.002 | 0.2138 ± 0.013 | 0.5907 |
| CARE-G | 0.6313 ± 0.002 | 0.8645 ± 0.004 | 0.8639 ± 0.003 | 0.4996 ± 0.001 | 0.7148 |
| CARE-InfoNCE | 0.7685 ± 0.001 | 0.7737 ± 0.006 | 0.8560 ± 0.005 | 0.8452 ± 0.032 | 0.8108 |
| CARE-WSC | 0.7639 ± 0.011 | 0.8019 ± 0.004 | 0.8538 ± 0.006 | 0.8581 ± 0.005 | 0.8194 |

(i.e., CARE-WSC) results in a 4% to 5% performance reduction on average. This aligns with the theoretical analysis in Theorem 3.4, suggesting that neither the vanilla nor weakly supervised contrastive losses are as effective in mitigating bias as the similarity-guided approach.

These results validate the necessity of the global affinity information and the similarity-guided contrastive loss, both of which are essential for the robustness and effectiveness of CARE.

## 4.4 Parameter Analysis

In this subsection, we delve into the parameter sensitivity analysis of the CARE framework on the four datasets, specifically exploring the impact of $\alpha$, $\lambda$, and the number of clusters. In the experiment, the mean and standard deviation of AUROC over five rounds are reported. We fix the number of clusters to be 10 and vary the values of $\alpha$ and $\lambda$ while recording the AUROC of CARE. The performance is visualized in Figure 2, where the x, y, and z axes represent $\alpha$, $\log(\lambda)$, and AUROC, respectively. From observation, CARE achieves the best performance when $\alpha = 0.4$ on the Amazon and DBLP datasets, while it reaches its peak performance at $\alpha = 0.01$ on the BlogCatalog dataset and $\alpha = 1$ on the IMDB dataset. Upon investigation, we find

that the number of edges in the first view on the IMDB dataset is unusually large compared to the number of edges in the second view as presented in Table 2. Due to this unusual pattern, we hypothesize that the raw adjacency matrix is not reliable and thus the global node affinity (i. e., similarity of soft assignment) plays a crucial role in detecting anomalous nodes. Thus, by setting $\alpha = 1$, we exclude the local node affinity in the loss function $\mathcal{L}_A$ and therefore CARE achieves better performance solely relying on the global node affinity. In terms of the parameter analysis on $\lambda$, we find that CARE prefers a smaller value of $\lambda$ (e. g., $\lambda = 0.1$). One explanation for this is that a large value of $\lambda$ tends to overly dominate the optimization of the overall objective function as $\lambda$ is used to balance the importance between the similarity-guided contrastive loss and the node affinity learning loss. In the subsequent experiment, we scrutinize the role of the number of clusters in shaping the anomaly detection criteria. Figure 3 illustrates the results, with the x and y axes representing the values of the number of clusters and AUROC, respectively. We observe that changing the number of the clusters does not greatly influence the performance of CARE across these four datasets. Notably, CARE achieves better performance when the number of clusters is 5 or 10 on most datasets, suggesting that the

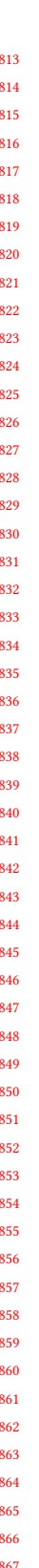

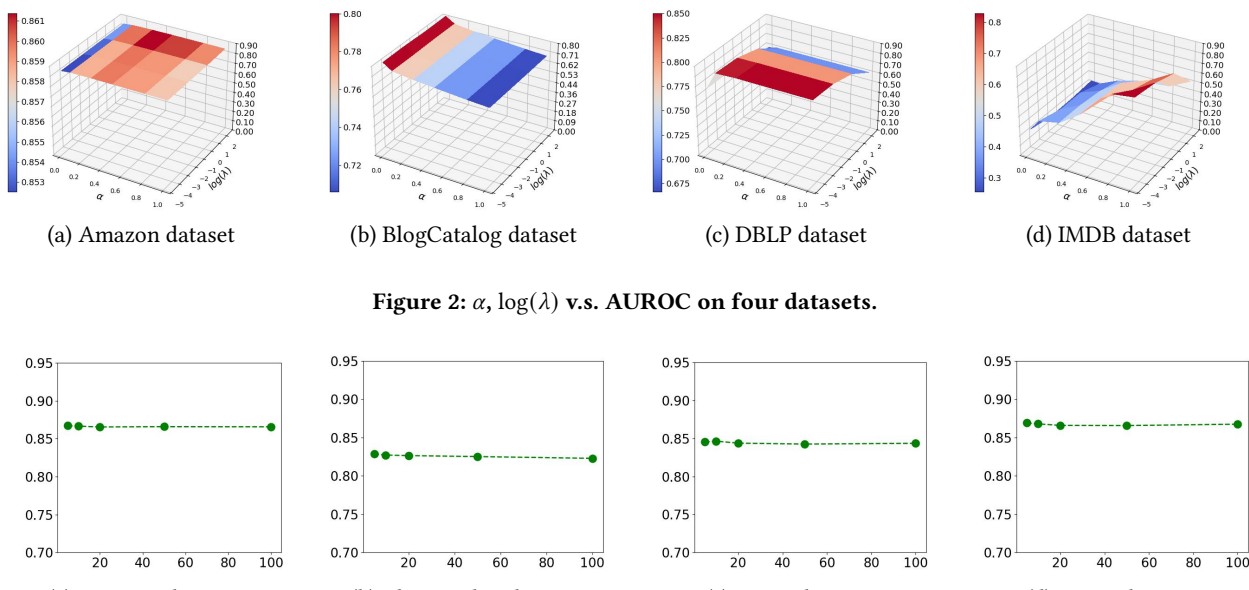

(a) Amazon dataset      (b) BlogCatalog dataset      (c) DBLP dataset      (d) IMDB dataset

**Figure 2:** $\alpha$, $\log(\lambda)$ **v.s. AUROC on four datasets.**

(a) Amazon dataset      (b) BlogCatalog dataset      (c) DBLP dataset      (d) IMDB dataset

**Figure 3: The number of clusters v.s. AUROC on four datasets.**

proposed method prefers a small number of clusters. By reducing the number of clusters, our method tends to generalize better by capturing larger, more meaningful patterns in the data instead of overfitting to noise.

## 4.5 Efficiency Analysis

**Time Complexity.** In this subsection, we analyze the time complexity of our proposed CARE. Assume that the graph has $n$ nodes, the input feature dimension is $d$, the hidden feature dimension is $f$, the number of nodes is $n$, the number of edges is $|E|$, and the number of clusters is $k$. For ease of explanation, we only consider the 1-layer case. Following [7], the time complexity of computing the soft membership matrix using GCN is $O(ndk + n|E|k + n)$. The complexity of the GCN to capture the hidden representation $h$ is $O(|E|d + ndf)$ with sparse computation. The complexity of the similarity-guided contrastive learning loss is $O(n^2 f)$. However, in the experiments, we can use the sampling strategy to sample $p(p << n)$ nodes and thus the complexity can be reduced to $O(p^2 f)$. The total complexity of CARE is $O(n(kd + fd + |E|k + 1) + |E|d + p^2 f)$.

**Running Time Analysis.** Next, we experiment on the YelpChi dataset to show the efficiency of our proposed CARE by changing the number of nodes and the number of layers. The reason why we only provide the efficiency analysis regarding the training/running time vs the number of nodes on the YelpChi dataset is that we can manually increase the number of nodes from 500 to 15,000 on this dataset. For the other datasets (e.g., IMDB, BlogCatalog, Amazon, DBLP), the number of nodes is less than 15,000 (see Table 2 for details) and we cannot get the running time if the number of nodes is set to be a value larger than 15,000. Thus, we do not provide the efficiency analysis on small datasets. In the first experiment, we fix the number of layers to be 1, the total number of iterations to be 10,000 and adjust the number of nodes by randomly selecting

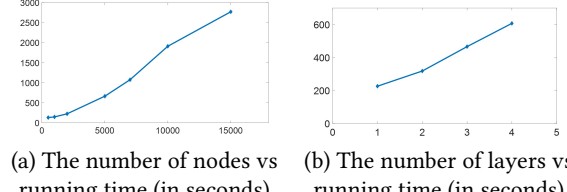

(a) The number of nodes vs running time (in seconds)      (b) The number of layers vs running time (in seconds)

**Figure 4: Efficiency analysis on the YelpChi dataset**

$k$ nodes, where $k \in [500, 1000, 2000, 5000, 10000, 15000]$. The experimental result is shown in Figure 4 (a). By observation, we find that the running time is quadratic to the number of nodes. In the second experiment, we fixed the number of nodes to be 2000, the total number of iterations to be 10,000, and adjusted the number of GCN layers from 1 layer to 4 layers. The experimental result is shown in Figure 4 (b). We observe that the running time is almost linear with respect to the number of layers.

## 5 Conclusion

In this paper, we proposed a novel self-supervised framework for anomaly detection in multi-view graphs, addressing key limitations of existing methods. By capturing both local and global node affinities and leveraging a similarity-guided contrastive loss, our approach effectively identifies anomalous nodes without relying on strong structural assumptions. The proposed method not only augments the graph structure with learned soft membership assignments but also mitigates the negative impact of low-quality assignments through a robust loss function, theoretically connected to graph spectral clustering. Our experimental results on six datasets demonstrate the superior performance of the proposed framework in detecting anomalies in complex, multi-view graphs.

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

# A Proof

## A.1 Proof for Lemma 3.1

**Lemma 3.1:** *(Similarity-guided Graph Contrastive Loss) Let $\bar{M}$ be the output of a one-layer graph neural network defined in Eq. 3. Then, $\mathcal{L}_C$ is equivalent to the following loss function:*

$$\mathcal{L}_C = \mathcal{L}_f + C$$

*where $\mathcal{L}_f = -\sum_{i=1}^n \sum_{j=1}^n \log \frac{\exp(2\tilde{A}_{ij}\bar{h}_i\bar{h}_j^T)}{\Pi_{k=1}^n \exp((\bar{h}_i\bar{h}_k^T)^2)^{1/n}}$ is a graph contrastive loss and $C$ is a constant.*

Proof.

$$\min_{\bar{H}} \mathcal{L}_C = \min_{\bar{H}} ||\tilde{A} - \bar{H}\bar{H}^T||_F^2$$

$$= \min_{\bar{H}} \sum_{i=1}^n \sum_{j=1}^n (\tilde{A}_{ij} - \bar{h}_i\bar{h}_j^T)^2$$

$$= \min_{\bar{H}} \sum_{i=1}^n \sum_{j=1}^n (\tilde{A}_{ij}^2 - 2\tilde{A}_{ij}\bar{h}_i\bar{h}_j^T + (\bar{h}_i\bar{h}_j^T)^2)$$

Notice that $\bar{M}$ is independent of $\bar{H}$. When we fix the parameter $\bar{M}$ to update $\bar{H}$, then $\tilde{A}_{ij}^2$ can be considered as a constant in this optimization problem. Thus, we have

$$\min_{\bar{H}} \mathcal{L}_C = \min_{\bar{H}} \sum_{i=1}^n \sum_{j=1}^n (-2\tilde{A}_{ij}\bar{h}_i\bar{h}_j^T + (\bar{h}_i\bar{h}_j^T)^2) + C$$

$$= \min_{\bar{H}} \sum_{i=1}^n \sum_{j=1}^n (-2\tilde{A}_{ij}\bar{h}_i\bar{h}_j^T + \frac{1}{n}\sum_{k=1}^n (\bar{h}_i\bar{h}_k^T)^2) + C$$

$$= \min_{\bar{H}} -\sum_{i=1}^n \sum_{j=1}^n \log \frac{\exp(2\tilde{A}_{ij}\bar{h}_i\bar{h}_j^T)}{\Pi_{k=1}^n \exp((\bar{h}_i\bar{h}_k^T)^2)^{\frac{1}{n}}} + C$$

$$= \min_{\bar{H}} \mathcal{L}_f + C$$

where $\mathcal{L}_f = -\sum_{i=1}^n \sum_{j=1}^n \log \frac{\exp(2\tilde{A}_{ij}\bar{h}_i\bar{h}_j^T)}{\Pi_{k=1}^n \exp((\bar{h}_i\bar{h}_k^T)^2)^{1/n}}$. Thus, we have $\mathcal{L}_C = \mathcal{L}_f + C$, which completes the proof. □

## A.2 Proof for Lemma 3.2

**Lemma 3.2:** *(Graph Contrastive Spectral Clustering) Let $\bar{M}$ be the output of a one-layer graph neural network defined in Eq. 3 and $\bar{h}_i$ and $\bar{h}_j$ be unit vectors. Then, minimizing $\mathcal{L}_C$ is equivalent to minimizing the following loss function:*

$$\min \mathcal{L}_C = \min[2Tr(\bar{H}^T L\bar{H}) + R(\bar{H})]$$

*where $L = I - \tilde{A}$ can be considered as the normalized graph Laplacian, $I$ is the identity matrix and $R(\bar{H}) = ||\bar{H}\bar{H}^T||_F^2$ is the regularization term.*

Proof. Based on the proof in Lemma 3.1, we have

$$\min \mathcal{L}_C = \sum_{i=1}^n \sum_{j=1}^n (-2\tilde{A}_{ij}\bar{h}_i\bar{h}_j^T + (\bar{h}_i\bar{h}_j^T)^2)$$

$$= (\sum_{i=1}^n \sum_{j=1}^n (2\tilde{A}_{ij} - 2\tilde{A}_{ij} - 2\tilde{A}_{ij}\bar{h}_i\bar{h}_j^T + (\bar{h}_i\bar{h}_j^T)^2)$$

Notice that $\tilde{A} = \tilde{D}^{-1/2}(\alpha\bar{M}\bar{M}^T + (1-\alpha)\frac{1}{v}\sum_{a=1}^v A^a)\tilde{D}^{-1/2}$ and $\tilde{D}_{ii} = \sum_j (\alpha\bar{M}_i\bar{M}_j^T + (1-\alpha)\frac{1}{v}\sum_{a=1}^v A_{ij}^a)$. We have $\sum_{j=1}^n \tilde{A}_{ij} = 1$ and thus $\sum_{i=1}^n \sum_{j=1}^n 2\tilde{A}_{ij}$ is a constant, which can be ignored in this optimization problem. Since $\bar{h}_i$ and $\bar{h}_j$ are unit vectors, we have

$$\min \mathcal{L}_C = \sum_{i=1}^n \sum_{j=1}^n (2\tilde{A}_{ij} - 2\tilde{A}_{ij}\bar{h}_i\bar{h}_j^T + (\bar{h}_i\bar{h}_j^T)^2)$$

$$= \sum_{i=1}^n \sum_{j=1}^n \tilde{A}_{ij}||\bar{h}_i||_2^2 + \sum_{i=1}^n \sum_{j=1}^n \tilde{A}_{ij}||\bar{h}_j||_2^2 - \sum_{i=1}^n \sum_{j=1}^n 2\tilde{A}_{ij}\bar{h}_i\bar{h}_j^T$$

$$+ \sum_{i=1}^n \sum_{j=1}^n (\bar{h}_i\bar{h}_j^T)^2$$

$$= \sum_{i=1}^n I_{ii}||\bar{h}_i||_2^2 + \sum_{j=1}^n I_{jj}||\bar{h}_j||_2^2 - \sum_{i=1}^n \sum_{j=1}^n 2\tilde{A}_{ij}\bar{h}_i\bar{h}_j^T$$

$$+ \sum_{i=1}^n \sum_{j=1}^n (\bar{h}_i\bar{h}_j^T)^2$$

$$= 2Tr(\bar{H}^T L\bar{H}) + R(\bar{H})$$

(13)

where $L = I - \tilde{A}$ can be considered as the normalized graph Laplacian, $I$ is the identity matrix and $R(\bar{H}) = ||\bar{H}\bar{H}^T||_F^2$, which completes the proof. □

## A.3 Proof for theorem 3.4

**Definition 3.3** *Given a sample $x_i$, we say $(x_i, x_j)$ is a false negative pair (or a true positive pair), if their optimal representations satisfy $\exp(\bar{h}_i\bar{h}_j^T/\tau) > 1$ for a small positive value $\tau$. Similarly, we say $(x_i, x_k)$ is a true negative pair (or a false positive pair), if their optimal representations satisfy $\exp(\bar{h}_i\bar{h}_k^T/\tau) \approx 0$ for a small positive value $\tau$.*

**Theorem 3.4** *Given the contrastive learning loss function $\mathcal{L}_3$, if there exists one false positive sample in the batch during training, the contrastive learning loss will lead to a sub-optimal solution.*

Proof. We can rewrite $\mathcal{L}_3$ as follows:

$$\mathcal{L}_3 = \sum_i \sum_{j \in C(i), j \neq i} [\log(\frac{\exp(\bar{h}_i\bar{h}_j^T/\tau) + \sum_{k \notin C(i)} \exp(\bar{h}_i\bar{h}_k^T/\tau)}{\exp(\bar{h}_i\bar{h}_j^T/\tau)})]$$

$$= \sum_i \sum_{j \in C(i), j \neq i} [\log(\exp(\bar{h}_i\bar{h}_j^T/\tau) + \sum_{k \notin C(i)} \exp(\bar{h}_i\bar{h}_k^T/\tau)) - \bar{h}_i\bar{h}_j^T/\tau]$$

$$= \sum_{j \in C(1), j \neq 1} [\log(\exp(\bar{h}_1\bar{h}_j^T/\tau) + \sum_{k \notin C(1)} \exp(\bar{h}_1\bar{h}_k^T/\tau)) - \bar{h}_1\bar{h}_j^T/\tau]$$

$$+ \sum_{i \neq 1} \sum_{j \in C(i), j \neq i} [\log(\exp(\bar{h}_i\bar{h}_j^T/\tau) + \sum_{k \notin C(i)} \exp(\bar{h}_i\bar{h}_k^T/\tau)) - \bar{h}_i\bar{h}_j^T/\tau]$$

(14)

Here, we select $x_1$ as the anchor node such that $(x_i, x_1)$ is a true positive pair (i. e., $x_i$ and $x_1$ are from the same cluster). Taking the derivative of $\mathcal{L}_3$ with respect to $\bar{h}_1$, we have

$$\frac{\partial \mathcal{L}_3}{\partial \bar{h}_1} = \frac{1}{\tau} \sum_{j \in C(1), j \neq 1} \left[ \frac{\exp(\bar{h}_1 \bar{h}_j^T / \tau) \bar{h}_j + \sum_{k \notin C(1)} \exp(\bar{h}_1 \bar{h}_k^T / \tau) \bar{h}_k}{\exp(\bar{h}_1 \bar{h}_j^T / \tau) + \sum_{k \notin C(1)} \exp(\bar{h}_1 \bar{h}_k^T / \tau)} - \bar{h}_j \right]$$

$$+ \frac{1}{\tau} \sum_{i \neq 1, i \in C(1)} \left[ \frac{\exp(\bar{h}_i \bar{h}_1^T / \tau) \bar{h}_i}{\exp(\bar{h}_i \bar{h}_1^T / \tau) + \sum_{k \notin C(i)} \exp(\bar{h}_i \bar{h}_k^T / \tau)} - \bar{h}_i \right]$$

$$= \frac{1}{\tau} \sum_{j \in C(1), j \neq 1} \left[ \frac{\sum_{k \notin C(1)} \exp(\bar{h}_1 \bar{h}_k^T / \tau) \bar{h}_k - \sum_{k \notin C(1)} \exp(\bar{h}_1 \bar{h}_k^T / \tau) \bar{h}_j}{\exp(\bar{h}_1 \bar{h}_j^T / \tau) + \sum_{k \notin C(1)} \exp(\bar{h}_1 \bar{h}_k^T / \tau)} \right]$$

$$+ \frac{1}{\tau} \sum_{i \neq 1, i \in C(1)} \left[ \frac{- \sum_{k \notin C(i)} \exp(\bar{h}_i \bar{h}_k^T / \tau) \bar{h}_i}{\exp(\bar{h}_i \bar{h}_1^T / \tau) + \sum_{k \notin C(i)} \exp(\bar{h}_i \bar{h}_k^T / \tau)} \right]$$

Setting the gradient to 0, we have

$$\sum_{j \in C(1), j \neq 1} \left[ \frac{\sum_{k \notin C(1)} [\exp(\bar{h}_1 \bar{h}_k^T / \tau) \bar{h}_j - \exp(\bar{h}_1 \bar{h}_k^T / \tau) \bar{h}_k]}{\exp(\bar{h}_1 \bar{h}_j^T / \tau) + \sum_{k \notin C(1)} \exp(\bar{h}_1 \bar{h}_k^T / \tau)} \right]$$

$$+ \sum_{i \neq 1, i \in C(1)} \left[ \frac{\sum_{k \notin C(i)} \exp(\bar{h}_i \bar{h}_k^T / \tau) \bar{h}_i}{\exp(\bar{h}_i \bar{h}_1^T / \tau) + \sum_{k \notin C(i)} \exp(\bar{h}_i \bar{h}_k^T / \tau)} \right] = 0 \quad (15)$$

As $(x_i, x_1)$ is a true positive pair, both $(x_i, x_k)$ and $(x_1, x_k)$ are true negative pairs. According to Definition 3.3, $\exp(\bar{h}_i \bar{h}_k^T / \tau) \approx 0$ and $\exp(\bar{h}_1 \bar{h}_k^T / \tau) \approx 0$ for some positive small values $\tau$ and both two terms of Eq. 15 is 0. Thus, Eq. 15 holds.

Next, we want to show that if there exists one false positive sample, we reach a contradiction. Assuming that $x_1$ is one false positive sample of $x_i$ in the batch, then both $(x_1, x_j)$ and $(x_i, x_1)$ are false positive pairs and $(x_1, x_k)$ is a false negative pair for some $k$ (e.g., $x_1$ and $x_k$ are from the same cluster for some $k$). Similarly, as $(x_i, x_k)$ is a true negative pair for all $k$, $\exp(\bar{h}_i \bar{h}_k^T / \tau) \approx 0$ and the second term of Eq. 15 is approximately 0. Therefore, we have

$$\sum_{j \in C(1), j \neq 1} \frac{\sum_{k \notin C(1)} [\exp(\bar{h}_1 \bar{h}_k^T / \tau) \bar{h}_j - \exp(\bar{h}_1 \bar{h}_k^T / \tau) \bar{h}_k]}{\exp(\bar{h}_1 \bar{h}_j^T / \tau) + \sum_{k \notin C(1)} \exp(\bar{h}_1 \bar{h}_k^T / \tau)} + 0 = 0$$

$$\sum_{j \in C(1), j \neq 1} \sum_{k \notin C(1)} [\exp(\bar{h}_1 \bar{h}_k^T / \tau) \bar{h}_j - \exp(\bar{h}_1 \bar{h}_k^T / \tau) \bar{h}_k] = 0 \quad (16)$$

We multiply Eq. 16 by $\bar{h}_i^T$, where $(x_i, x_j)$ is a true positive pair for any $j$ (i.e., both $x_i$ and $x_j$ are from the same cluster) and $(x_i, x_k)$ is a true negative pair for any $k$. Then, we have

$$\sum_{j \in C(1), j \neq 1} \sum_{k \notin C(1)} [\exp(\bar{h}_1 \bar{h}_k^T / \tau) \bar{h}_j \bar{h}_i^T$$

$$+ \exp(\bar{h}_1 \bar{h}_k^T / \tau)(-\bar{h}_k \bar{h}_i^T)] = 0 \quad (17)$$

Since $(x_i, x_j)$ is a true positive pair and $(x_i, x_k)$ is a true negative pair, we have $\exp(\bar{h}_j \bar{h}_i^T / \tau) > 1$ and $\exp(\bar{h}_k \bar{h}_i^T / \tau) \approx 0$, which means that

$\bar{h}_j \bar{h}_i^T > 0$ and $\bar{h}_k \bar{h}_i^T < 0$. Therefore, both two terms of Eq. 17 are non-negative and Eq. 17 holds if and only if $\exp(\bar{h}_1 \bar{h}_k^T / \tau) \approx 0$ for any $k \notin C(1)$ (i.e., $(x_1, x_k)$ is a true negative pair for any $k \notin C(1)$).

If $(x_1, x_k)$ is a true negative pair for any $k \notin C(1)$, then $(x_1, x_i)$ has to be a true positive pair, and it contradicts our assumption that $x_1$ is a false positive sample of $x_i$. Therefore, we reach a contradiction and we could not get the optimal solution for $\bar{h}_1$, which completes the proof. □

# B  Experiments

In this section, we show the details of generating anomalous node for the semi-supervised datasets, including IMDB and DBLP. We also show the hyper-parameter specification for reproducing the experimental results.

## B.1  Anomalous node generation

We follow the published works [14] to generate anomalous nodes by perturbing the topological structure or node attributes of an attributed network. To perturb the topological structure of an attributed network, we adopt the method introduced by [14] to generate some small cliques as in many real-world scenarios. A small clique is a typical anomalous substructure due to larger node degrees than normal nodes. By [1], after we specify the clique size as $m$, we randomly select $m$ nodes from the network and then make those nodes fully connected. Then all the $m$ nodes in the clique are regarded as anomalies. In addition to the injection of structural anomalies, we adopt another attribute perturbation schema introduced by [14] to generate anomalies from an attribute perspective. For each selected node $u_i$, we randomly pick another $k$ nodes and select node $u_j$ whose attributes deviate the most from node $u_j$ among the $k$ nodes by maximizing the Euclidean distance $||x_i - x_j||^2$. Afterward, we then change the attributes $x_i$ of node $u_i$ to $x_j$.

## B.2  Reproducibility

All of the real-world data sets are publicly available. The experiments are performed on a Windows machine with a 24GB RTX 4090 GPU. We use TAM as the backbone of our method to capture the local node affinity. We set the number of clusters to be 10, $\lambda = 0.1$ and $\alpha = 0.8$ for the CERT dataset. For the IMDB dataset, we set the number of clusters to be 10, the value of $\lambda = 1$ and $\alpha = 0.8$. For the DBLP dataset, we set the number of clusters to be 10, the value of $\lambda = 0.01$ and $\alpha = 0.8$. For the BlogCatalog dataset, we set the number of clusters to be 5, $\lambda = 0.01$ and $\alpha = 0.01$. For the Amazon dataset, we set the number of clusters to be 10, $\lambda = 1$ and $\alpha = 0.8$. For the Yelp dataset, we set the number of clusters to be 10, $\lambda = 1$ and $\alpha = 0.8$.

