# OpenReview forum: "Cluster Aware Graph Anomaly Detection"
_ACM.org/TheWebConf/2025/Conference — WWW 2025 Oral_

### Official Review · Reviewer_1dp9 · 2024-11-16

**Novelty:** 5
**Technical Quality:** 5

**Review:**

This paper presents CARE, a cluster-aware multi-view graph anomaly detection method that enhances adjacency matrices with pseudo-labels to capture node affinities and uses a similarity-guided loss to mitigate bias.  Experimental results highlight its superiority over existing methods.

**Strength:**
1. Performance: Extensive experiments and significant improvements demonstrate the effectiveness of the proposed method in graph anomaly detection tasks.

2. Theory: This paper systematically analyzes the potential negative impacts of traditional weakly supervised contrastive learning on the model and provides detailed theoretical derivations.

3. Method: CARE augments the graph's adjacency matrix with pseudo-labels and introduces a similarity-guided loss to mitigate potential biases from the pseudo-labels. The method is intuitive and easy to follow.

**Weakness:**

1. Experiment: Performance is sensitive to hyperparameters, such as the similarity balancing coefficient (𝛼) and regularization coefficient (𝜆). It is necessary to provide hyperparameter tests.

**Questions:**

See Section Weakness.

**Reviewer Confidence:**

2: The reviewer is willing to defend the evaluation, but it is likely that the reviewer did not understand parts of the paper

**Scope:**

4: The work is relevant to the Web and to the track, and is of broad interest to the community

---

### Official Review · Reviewer_nSeK · 2024-11-19

**Novelty:** 6
**Technical Quality:** 6

**Review:**

The manuscript proposes CARE, a cluster-aware multi-view graph anomaly detection framework. CARE augments the adjacency matrix with pseudo-labels, employs a similarity-guided contrastive loss, and connects these approaches to graph spectral clustering. Experimental results on diverse datasets demonstrate that CARE outperforms state-of-the-art methods in terms of AUROC and AUPRC.

### Strengths
- CARE introduces a novel method combining local and global affinities, leveraging pseudo-labels, and similarity-guided loss.
- Establishes the connection between the proposed similarity-guided contrastive loss and graph spectral clustering.
- Effective on both single-view and multi-view graphs, showcasing versatility.
- Provides code for reproducibility, enhancing its utility for the research community.


### Weaknesses
- The similarity-guided loss has \(O(n^2)\) complexity, which might become infeasible for large graphs despite the proposed sampling strategy.
- Experiments are limited to relatively small datasets (maximum ~25,000 nodes); results on larger-scale graphs are not explored.
- The manuscript overlooks several closely related studies that could enhance its theoretical and methodological comparisons: [1]. Multi-level hyperedge distillation for social linking prediction on sparsely observed networks - Proceedings of the Web Conference 2021.  [2]. Heterogeneous hypergraph embedding for graph classification - Proceedings of the 14th ACM International Conference on Web Search and Data Mining.   [3]. Graph masked autoencoders with transformers - arXiv preprint arXiv:2202.08391.  These works offer complementary perspectives on graph embedding and could strengthen the discussion on multi-view graph representations.
- The conclusion does not adequately detail future research avenues. It would benefit from including emerging topics, such as prompt learning (All in One: Multi-Task Prompting for Graph Neural Networks - KDD2023 ) and sparsification strategies ( Graph Sparsification via Mixture of Graphs - arXiv preprint.  Adaptive Coordinators and Prompts on Heterogeneous Graphs for Cross-Domain Recommendations - arXiv preprint.  ) that align with CARE's potential extensions.

**Questions:**

N/A

**Reviewer Confidence:**

3: The reviewer is confident but not certain that the evaluation is correct

**Scope:**

4: The work is relevant to the Web and to the track, and is of broad interest to the community

---

### Official Review · Reviewer_AT93 · 2024-11-26

**Novelty:** 5
**Technical Quality:** 4

**Review:**

This paper presents a new self-supervised framework for anomaly detection over multi-view graphs. To address the view heterogeneity and one-class homophily assumption, this work proposes a contrastive loss built on the augmented graph from the cluster soft membership assignments and the GCN-based node representations, which capture both local and global node affinity and achieve robust anomaly detection. Theoretical connections to spectral clustering are provided and empirical results show the effectiveness of the proposed approach.

Strong Points:
1. The paper is easy to follow.
2. Theoretical justifications are provided and explanations for the loss functions are detailed.
3. The empirical effectiveness is impressive, gaining a significant improvement of 39% and 18.7% in AUPRC/AUROC on Amazon and YelpChi, respectively.

Weak Points:
1. The major technical contributions lie on the loss functions.
2. The evaluated baselines are relatively weak. Several recent mult-view graph-based anomaly detection methods are not compared or discussed. It would be better to include more strong competitors for comparison.
- Duan et al. Graph Anomaly Detection via Multi-Scale Contrastive Learning Networks with Augmented View. AAAI'23
- Zhang et al. Reconstruction Enhanced Multi-View Contrastive Learning for Anomaly Detection on Attributed Networks. IJCAI'22
- Lian et al. Graph Anomaly Detection via Multi-View Discriminative Awareness Learning. TNSE.
3. The backbone of the proposed model is GCN. It would be better to report the model performance using other popular GNN models, e.g., SGC, APPNP, GAT, and GCNII.
4. In the proposed framework, all views are equally treated (in Eq. (3) and (4)), while existing works related to multi-view (graph) data all involve learning weights for different views. It would be better to explain this difference more clearly. Why the weighting for views is not important here? I am asking this question since this work is tailored for multi-view graphs but the proposed approach seems to be simply a single-view approach.

**Questions:**

1. In Lemmata 3.1 and 3.2, the authors have proved that optimizing the loss function in Eq. (7) is equivalent to minimizing $L_f+C$ (the minimization symbol is missed in Eq. (8)) or optimizing the graph Laplacian smoothing problem in Eq. (9) (there is a redundant minimization symbol before $L_C$ in Eq. (9)).  I am wondering if $L_C$ in Eq. (10) is replaced by these two losses, will the detection performance remain the same?
2. Weak Points 1-4.

**Reviewer Confidence:**

3: The reviewer is confident but not certain that the evaluation is correct

**Scope:**

3: The work is somewhat relevant to the Web and to the track, and is of narrow interest to a sub-community